# Xylella fastidiosa subsp. pauca and olive produced lipids moderate the switch adhesive versus non-adhesive state and viceversa

**Valeria Scala[1]\*, Nicoletta Pucci[1], Manuel Salustri[2], Vanessa Modesti[1], Alessia L'Aurora[1], Marco Scortichini[3], Marco Zaccaria[4], Babak Momeni[4], Massimo Reverberi[2]\*, Stefania Loreti[1]**

**1** Council for Agricultural research and Economics (CREA), Research Centre for Plant Protection and Certification, Roma, Italy, **2** Dept. of Environmental Biology, Sapienza University, Roma, Italy, **3** Council for Agricultural research and Economics (CREA), Research Centre for Olive, Fruit Trees and Citrus, Roma, Italy, **4** Department of Biology, Boston College, Chestnut Hill, MA, United States of America

\* massimo.reverberi@uniroma1.it (MR); valeria.scala@crea.gov.it (VS)

**Data Availability Statement:** All relevant data are within the manuscript and its Supporting Information files.

**Funding:** The study was funded by MIPAAFT, Project Oli.Di.X.I.It ("OLIvicoltura e Difesa da

## Abstract

Global trade and climate change are re-shaping the distribution map of pandemic pathogens. One major emerging concern is *Xylella fastidiosa*, a tropical bacterium recently introduced into Europe from America. In last decades, *X. fastidiosa* was detected in several European countries. *X. fastidiosa* is an insect vector-transmitted bacterial plant pathogen associated with severe diseases in a wide range of hosts. *X. fastidiosa* through a tight coordination of the adherent biofilm and the planktonic states, invades the host systemically. The planktonic phase is correlated to low cell density and vessel colonization. Increase in cell density triggers a quorum sensing system based on mixture of cis 2-enoic fatty acids—diffusible signalling factors (DSF) that promote stickiness and biofilm. The lipidome profile of *Olea europaea* L. (cv. Ogliarola salentina) samples, collected in groves located in infected zones and uninfected zones was performed. The untargeted analysis of the lipid profiles of Olive Quick Decline Syndrome (OQDS) positive (+) and negative (-) plants showed a clustering of OQDS+ plants apart from OQDS-. The targeted lipids profile of plants OQDS+ and OQDS- identified a shortlist of 10 lipids that increase their amount in OQDS+ and *X. fastidiosa* positive olive trees. These lipid entities, provided to *X. fastidiosa* subsp. *pauca* pure culture, impact on the dual phase, e.g. planktonic ↔ biofilm. This study provides novel insights on OQDS lipid hallmarks and on molecules that might modulate biofilm phase in *X. fastidiosa* subsp. *pauca*.

## Introduction

*Xylella fastidiosa* (Xf) is one of the top 10 plant pathogenic bacteria [1] and is the cause of an environmental emergency within the European Union (EU). Xf has infected a broad host range [2] of plant species; symptoms vary depending on the combination of the host plant and Xf strain [3]. At present, *X. fastidiosa* subsp. *pauca*, associated to the OQDS, and three other

Xylella fastidiosa e da Insetti vettori in Italia"), D.M.
23773 del 6/09/2017, Project SALVAOLIVI
("Salvaguardia e valorizzazione del patrimonio
olivicolo italiano con azioni di ricerca nel settore
della difesa fitosanitaria"), D.M. 33437 del 21-12-
201 and by Regione Puglia agreement: "Strategie
di controllo integrato per il contenimento di Xylella
fastidiosa in oliveti pugliesi ed analisi
epidemiologiche del complesso del disseccamento
rapido dell'olivo (CoDiRO). B.M. and M.Z. were
supported by a start-up fund from Boston College
and by an Award in Biomedical Excellence from the
Smith Family Foundation. The funders had no role
in study design, data collection and analysis,
decision to publish, or preparation of the
manuscript.

**Competing interests:** The authors have declared
that no competing interests exist.

subspecies, *fastidiosa*, *multiplex*, and *sandyi* have been identified in Europe [4–7]. Xf subsp. *pauca* was initially introduced from America into Southern Italy and recovered in olive trees [8,9]. This pathogen is associated to the OQDS which has caused losses up to €390 M of the national olive oil production, in the last three years in Italy [10].

Xf is an obligated vector-transmitted pathogen and a xylem-limited bacterium. Inter-host transmission is mediated by xylem sap-feeding insects. The biology of invasion has been described in the grapevine [11]. From the point of entry in grapevine, Xf moves along the xylem, attaches to its walls and, through a tight coordination of the adherent biofilm and the planktonic states, invades the host systemically [12,13]. The planktonic phase is correlated to low cell density and vessel colonization [14]. The colonization during the exploratory phase (planktonic) is correlated to low cell density and symptoms progression in grapevine. Increase in cell density triggers a quorum sensing system based on cis 2-enoic fatty acids—DSF that promote stickiness and biofilm formation [15]. Xylem vessels are occluded by the combined effect of bacterial biofilm and plant defences such as tyloses and formation of amorphous gels coating xylem vessel walls, causing characteristic symptoms such as leaf scorch [11,16]. DSF can modulate gene-expression in Xf [17,18]. DSF-mediated quorum sensing determines a) degradation of the pit membranes to enable cross-vessel diffusion; b) twitching motility of Xf cells; c) adhesion to the xylem surface and biofilm formation. The earlier stage of infection in Pierce disease, consists of evasion by the pathogen of the plant innate immune response and colonization of its vessels, *de facto* limiting opportunities for reacquisition by the feeding insect vectors [19]. At this stage, Xf is not detected by the plant as a biotic stress, but rather as an abiotic stress (drought and dehydration) [20]. At later stages, the biofilm-based phenotype, consisting of a high density of Xf cells, facilitates reacquisition by the vector and dissemination into other hosts. Only at this point, the plant recognizes the pathogen and mounts an immune response ineffective at preventing Xf colonization and symptoms [21,22]. DSF act as coordinators of this dual activity of Xf, allowing the switch from the early stage (planktonic endophytic lifestyle) into the later one (sessile insect-acquisition stage) [11,23]. Lipids appear to be central in this pathogen interplay between the two phenotypes. In *Pseudomonas aeruginosa*, other lipids, namely the oxylipins, act as hormones for controlling the switch among the different stages of bacterial lifestyle: planktonic, twitching, and biofilm. In *P. aeruginosa*, the oleic acid-derived oxylipins control the virulence in the host and function as autoinducers of a novel quorum sensing system mediating cell-to-cell communication in bacteria [24,25].

To provide a broader view of the lipids emerging in OQDS symptomatic trees infected by Xf, we analyzed the lipidomic profile of samples of 60-y.o. *Olea europaea* L. cv. Ogliarola salentina symptomatic and symptomless for the OQDS. The profile of 437 lipid compounds was assayed: 186 were found to be differentially accumulated in OQDS positive individuals of which 90 were further characterized and quantified by MS/MS spectrometry. Among these, we identified ten compounds, more abundant in Xf+ and OQDS+, that are novel hallmarks of Xf symptoms and infection of olive trees. Importantly, we found that these lipids modulate Xf lifestyle under *in vitro* conditions.

## Materials and methods

### Study site and sampling procedures

Sampling was carried out in the Apulia region, in October 2017, on 120 individuals of *Olea europaea* L. cv. Ogliarola salentina (60 years old), 60 showing OQDS symptoms (+) and 60 OQDS symptomless (-). The OQDS+ individuals were collected in an olive grove in Copertino in the infected area of the Lecce province (40°16'5.56" N 18°03'15.48" E); the OQDS- individuals were sampled in Grottaglie, Taranto province (40°32'12.98" N 17°26'14.03" E) in an area

regarded by the phytosanitary service of the Apulia region as still unaffected by the pathogen. Trees were identified as symptomatic or symptomless following the criteria previously reported [26]. Mature shoots from each OQDS+ tree was collected from sections of the canopy on branches that showed desiccation and dieback. The samples consisted of "1- or 2-year-old twigs (ca.0.5 cm in diameter) from which cuttings of 15–20 cm were prepared from the portions close but yet unaffected by the withering and desiccation phenomena" [26]. Xylem tissue was then recovered, after removing the bark, and processed. The same procedure was performed for each OQDS- tree. The obtained 120 samples were separately lyophilized. 12 pools of equal weight (1g) were generated from the lyophilized samples, namely:6 pools (each representing 10 individuals) from OQDS+ and 6 from OQDS-. For a comprehensive lipidomic analysis, the pools included the overall analytical complexity of our samples [27,28]. OQDS + and OQDS- samples were molecularly assayed via real-time PCR in technical triplicates [3,29] to verify the presence of Xf and thereon defined as Xf+ and Xf- samples (S1 Fig).

## Lipids analysis

Xylem tissue (1,0 gr) was recovered and lipids extraction and analysis were performed as previously reported [30]. Xf+ and Xf- samples were assayed with the internal reference standards tricosanoic acid, glyceryl tripalmitate d31, and 9-HODEd4. The analysis was carried out at a final concentration of 2μM. The samples were analysed by untargeted lipid analysis conducted with a G6220A TOF-MS, (Agilent Technologies, USA) operating in negative and positive ion scan mode as previously described [30]. A sub-group of lipid classes was analysed (fragmentation analysis) by LC-MS/MS (Triple Quadrupole; 6420 Agilent Technologies, USA) as reported [30]; multiple reaction monitoring (MRM) methods were adopted to analyse the most abundant lipid entities (S1 Table). MRM data were processed using the Mass Hunter Quantitative software (B.07.00 version, Agilent Technologies, USA). The mass spectrometry analyses were performed three times, each time in technical triplicate (n = 9). PCA and significance-fold change analysis (Volcano plot) for untargeted LC-TOF/MS results were performed trough Agilent Mass Analyzer software. Significance tests (T-Student Test, $p < 0.05$) and plots of MRM and SIM results were performed trough R software.

## Quantification of biofilm formation

An *in vitro* test was made to assess the effect of free fatty acids, diacylglycerides, and oxylipins on biofilm formation of *Xylella fastidiosa* subsp. *pauca* strain De Donno (CFBP 8402). Xf subsp. *pauca* biofilm formation was evaluated as previously described [31] with some modifications. Briefly, a pure culture of the bacteria was grown for 7 days on PD2 plates, scraped and resuspended in PBS. 10μL of cell suspension (A600 = 0.5 OD) was inoculated in a sterile glass tube containing 1mL of PD2. The free fatty acids (FFA) (Sigma-Aldrich, USA) or diacylglycerides (1,3-Dilinoleoyl-rac-glycerol; 1,3-Diolein) (Sigma-Aldrich, USA) or oxylipins [Cayman chemicals, USA or 7,10 DiHOME and the mix (7,10 DiHOME; 10-HOME) kindly provided by Dr. Eriel Martínez and Javier Campos-Gómez (Southern Research Center, AL, USA] were added to the medium at desired concentrations when required as reported [24]. After 11 days of incubation (28˚C; 100 rpm), the total number of cells—planktonic growth (cells in suspension) and biofilm growth (cells adhered to the substrate) was estimated. The lipid compounds were divided into two groups: those dissolved in EtOH (with EtOH) and those dissolved in water (no EtOH). Spectrophotometric absorption of Xf subsp. *pauca* cultures was used to measure growth (600 nm) and biofilm formation (595 nm). For compounds dissolved in EtOH, references with a corresponding concentration of EtOH but without the compound were used as background. Impact on growth / impact on biofilm are defined as the amount of growth/

biofilm formation minus the background, normalized to the growth/biofilm formation in a medium in the absence of added EtOH or lipids. A positive impact indicates values more than no-lipid controls (i.e. improved growth or biofilming), whereas a negative impact indicates values less than no-lipid control (i.e. inhibited growth or biofilming). EtOH at similar molarity to lipids was additionally used as a point of reference of how Xf subsp. *pauca* growth and biofilms were affected. The experiments were performed in biological triplicate for each treatment and carried out three times (total repetitions per treatment n = 9). Multiple comparison with Kruskal-Wallis test (p value< 0.05) and Fisher's LSD post-hoc test, with Bonferroni correction, were run on R software to individuate significant (p<0.05) groupings within the different treatments.

## Results

### Infected trees exhibit a lipid profile that is distinct from healthy trees

A lipidomic untargeted analysis was performed to establish if lipids are differentially accumulated in OQDS symptomatic *versus* symptomless samples. OQDS + and OQDS- samples resulted respectively infected and non-infected with Xf when assessed by real-time PCR. Volcano plot of LC-ToF untargeted analyses on negative-ion scans identified 437 compounds, 186 of which significantly modified (Fold Change >2.0; p <0.05), with 173 upregulated and 13 downregulated (Fig 1). PCA highlighted two primary components (X-axis: 60.48%; Y-axis: 12.05%) that separated the Xf+ and Xf- clusters (Fig 2). Volcano plot analysis on positive-ion scans provided a few significantly modified compounds (S2 Fig).

Among the 186 differentially accumulated, 90 compounds were selected and analysed by a targeted MRM or SIM on the basis of the highest FC and in light of previous results [30] (S1 and S2 Tables). The chemometric analysis highlighted that ten compounds discriminated Xf + from Xf- samples and were significantly more abundant in Xf+ samples. Namely: oleic/linoleic/linolenic acid-deriving oxylipins (9-hydroxyoctadecenoic acid - 9HODE; 9-hydroperoxyoctatrienoic acid - 9HOTrE; 13-hydroxyoctadecenoic acid - 13HODE; 13-hydroperoxyoctatrienoic acid - 13HOTrE; 13-oxo-octadecenoic acid - 13oxoODE; 10-hydroxyoctadecenoic acid -10HODE; 10-hydroperoxyoctamonoenoic acid - 10HpOME); unsaturated fatty acids (oleic acid—C18:1; linoleic acid—C18:2); and diacylglycerol [DAG36:4 (18:1/18:3)] (Fig 3 and S3 Table).

### Lipids moderate Xf biofilm formation

In order to determine the effect of the compounds showed in Fig 3 on Xf ability to form biofilm, we proceeded to an *in vitro* test as reported in [24]. *In vitro* test of biofilm formation indicated that the 7,10-dihydroxyoctamonoenoic acid (7,10-diHOME), the mix 7.10 diHOME and 10-hydroxyoctamonoenoic acid (10-HOME), C18:1 and C18:2 and their diacylglycerols (DAG 36:2 and 36:4), at the concentration of 0.0025 mg mL$^{-1}$, stimulated planktonic growth (Fig 4A); at the same concentrations, 9-HODE induced biofilm formation, whereas 7,10-diHOME and the mix of 7,10 diHOME and 10-HOME strongly inhibited it (Fig 4B). Free fatty acids (linoleic acid, oleic acid) and diacylglycerides (1,3-Dilinoleoyl-rac-glycerol; 1,3-Dioleyl-rac-glycerol), tested at different concentrations, stimulated planktonic growth and did not affect biofilm formation (S3 and S4 Figs). LOX-derived oxylipins, namely 9HODE, 13HODE, 13OXODE and 9HOTRE promoted biofilm whilst the JA-related 13-HOTRE did not significantly affect it (S5 Fig). The DOX-derived oxylipins, 7,10-DiHOME and 10-HOME, produced by the pathogen (as suggested in [30]), strongly inhibited biofilm formation (S4 Fig). LOX-oxylipins did not show an effect on bacterial growth (S6 Fig), except for the DOX-oxylipins mixture of 7,10-DiHOME and 10-HOME (S3 Fig).

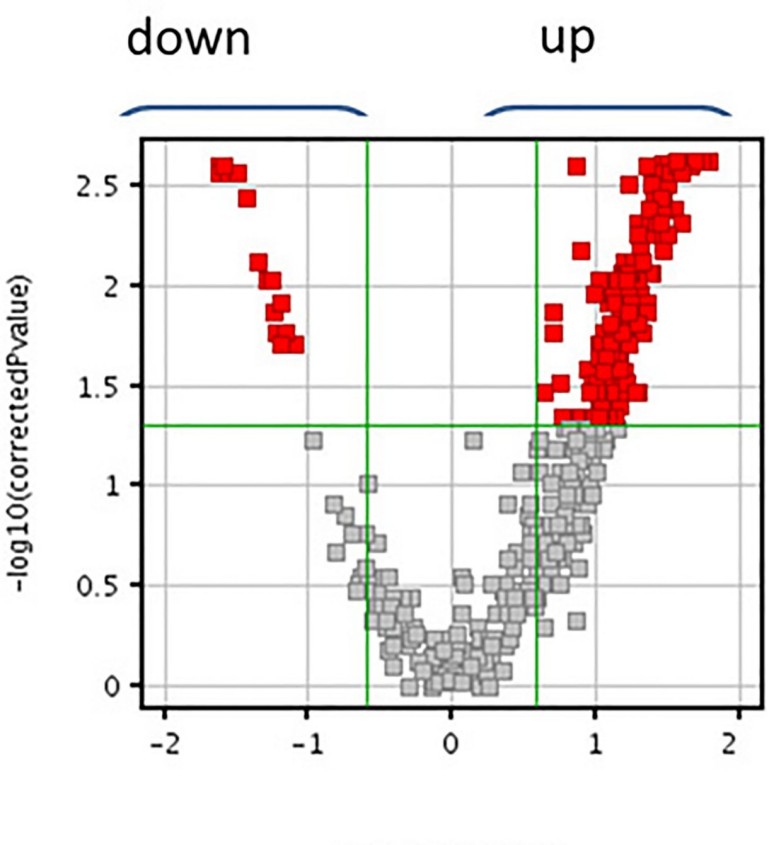

**Fig 1. Volcano plot analysis of 437 compounds on negative ion scan, in Xf+ and Xf- samples.** The x-axis shows the fold change to indicate the variation in abundance of compounds present in Xf+ compared to Xf- samples. The compounds on the right side were more abundant, whereas those on the left side were less abundant in Xf+ condition. Entities that satisfied the fold change and the *P*-value cut-off of 2.0 and 0.05, respectively, are marked in red.

## Discussion

In the host plant, Xf colonizes the xylem vessels and elicits their occlusion [11,23,32]. Xf maintains a lifestyle that switches bidirectionally from adhesive to non-adhesive cells phenotype and *vice versa* [14]. Lipids are crucial signals that can modulate the pathogen virulence and play important roles during the infection process [15,33]. Among lipids, oxylipins can modulate biofilm formation and virulence in the Gram-negative pathogen *P. aeruginosa* [25]. Most information on the biology of Xf is mostly referred to one of its subspecies, i.e. *fastidiosa*, in the context of Pierce's disease and, regarding lipids involvement, to DSF [21]. Recently, we showed that during the infection of the model organism *Nicotiana tabacum* by Xf subsp. *pauca*, several lipids were differently accumulated in infected *versus* healthy plants; DSF-like compounds apart, oxylipins emerged as hallmarks of pathogenic invasion of host tissues [30]. In olive trees, reports are available on the molecular basis of Xf subsp. *pauca* invasion [26,32]. More recently, Cardinale and colleagues [34] demonstrated that Xf subsp. *pauca* forms biofilm in xylem vessels of OQDS+ trees (cultivar "Ogliarola di Lecce"), pinpointing the role of bacterial aggregates in vessel occlusions in naturally infected olive trees. Notwithstanding the importance of lipids in Xf lifestyle, no report employs a lipidomic approach to differentiate OQDS+ from OQDS- trees.

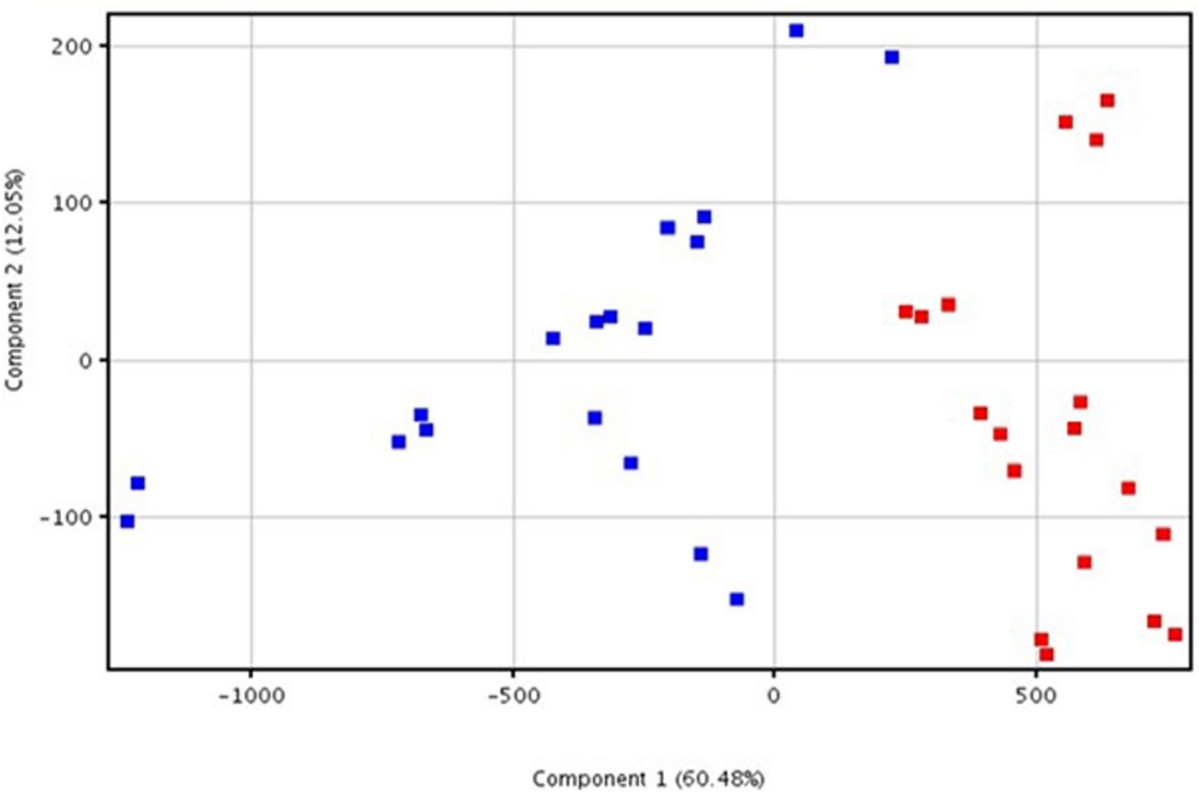

**Fig 2. Principal component analysis.** Score plot of data generated by HPLC-ESI/TOF-MS analysis of Xf+ (red square) and Xf- (blue square) samples. The results of the analysis referred to three separate experiments performed in triplicates.

We explored for the first time the lipidomic profile of olive trees naturally infected by *Xylella fastidiosa*. Lipids play a key role in plant disease. In this work, we check by untargeted analysis all the lipids that change under the pathogen pressure; moreover, PCA of the lipid entities allowed to differentiate the infected trees from the uninfected ones. Since the OQDS- samples clustered differently from OQDS+ samples, we suggest that the differential formation of lipid entities can be modulated by the presence of Xf producing OQDS symptoms. Lipids are essential constituents of the cells involved in different biological functions; this study could pave the way for developing a diagnostic tool targeted on the lipids that differentiate the infected plants by non-infected ones. In this regard, the Volcano plot analysis of the extracted entities from olive trees affected by OQDS, highlighted that 186 lipids compounds were specifically formed in symptomatic samples. More intriguingly, 10 compounds emerged as statistically able to discriminate Xf+ from Xf- olive trees. Considering that this part of the study gathers data straight from natural infection processes, we suggest these specific lipids as hallmarks of Xf infection of olive trees and propose to extend this type of approach to other hosts affected by this fastidious bacterial pathogen.

In olive trees, OQDS symptoms are closely related to xylem vessel occlusions caused by biofilm formation from Xf [34]. Lipid metabolism has a strong impact on plant-host interaction and, despite the proceedings in plant pathology, many important questions remain unanswered about the functional diversity of lipids and the mechanisms bacterial cells employ to coordinate their response to lipids modulation. In this study, we investigated how lipids differently accumulated in OQDS+ samples affected the lifestyle of *X. fastidiosa* subsp. *pauca*. To

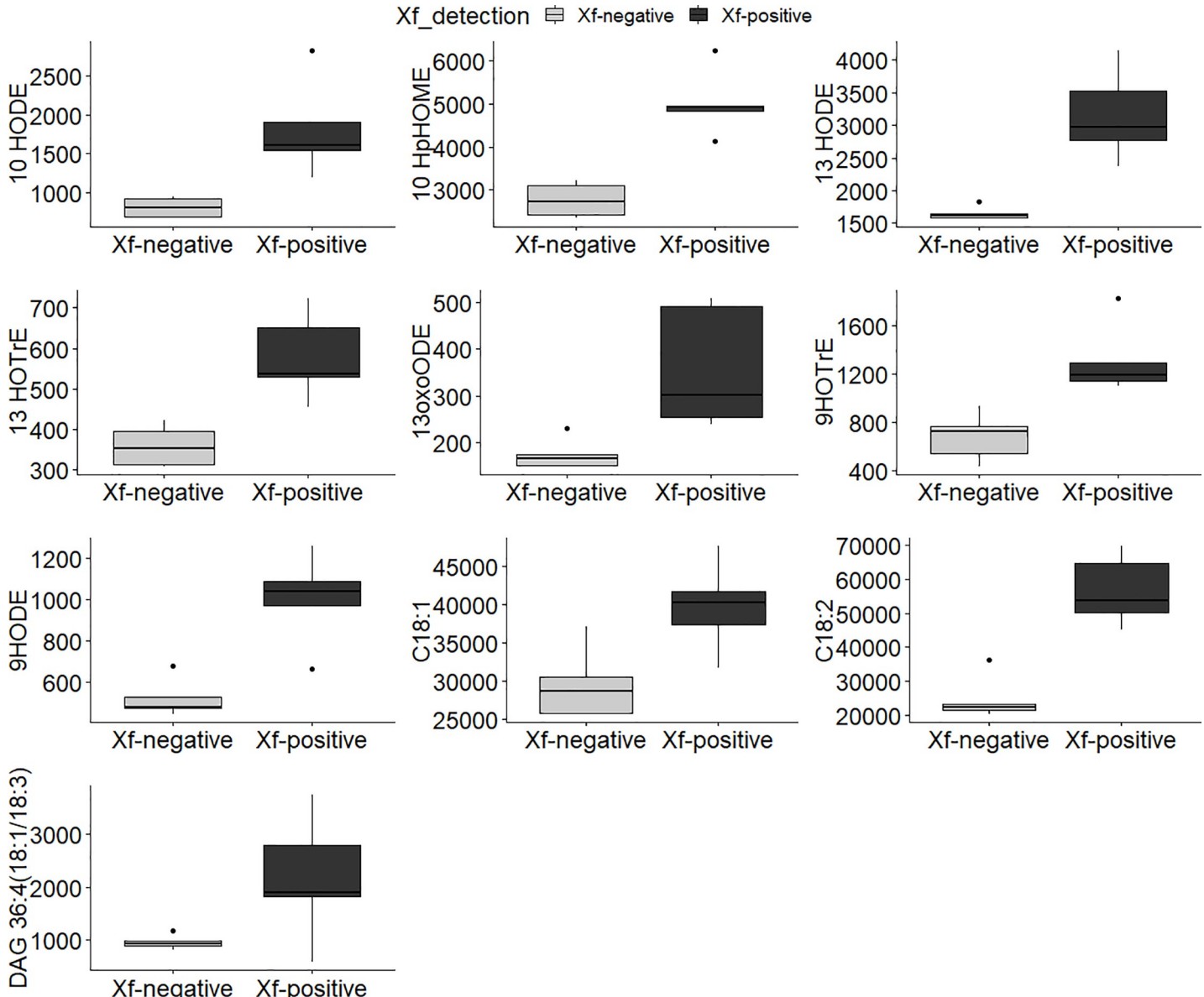

**Fig 3. Boxplots of statistically significant compounds analyzed by MRM or SIM.** Comparisons between naturally non-infected (Xf negative) and naturally infected (Xf positive) plant samples are displayed on X-axis. The Y-axis shows the normalized relative abundance. Horizontal line in each boxplot indicates the median and black dots represent the outlier samples.

identify their role in Xf biofilming [18,25], we tested *in vitro* the oleic acid-derived 7,10-diHOME alone as well as in conjunction with 10-HOME (DOX-oxylipins); the free fatty acids C18:1 and C18:2 and their diacylglycerols (DAG 36:2 and 36:4); the LOX-derived 9-oxylipins (9HODE, 9-OXODE, 9-HOTRE) and 13-oxylipins (13-HODE, 13OXODE and 13-HOTRE).

Our working hypothesis on the effects of these compounds on Xf biofilming stemmed from the available information on their biological role and was inspired by Roper and colleagues' intuition: "the bacteria induce an autoimmune-like syndrome" [14]. More specifically, the oleic acid is among the modulators of quorum sensing in Xf [15]; in other bacterial pathogens

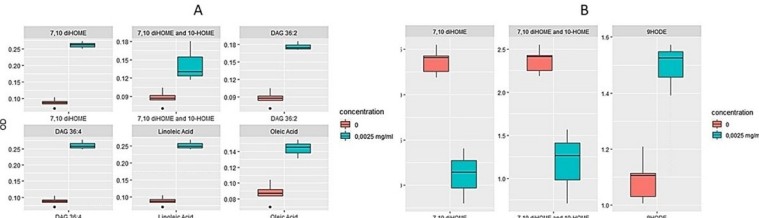

**Fig 4.** Boxplots of effects caused by lipid compounds on planktonic growth (A) and biofilm formation (B). Selected compounds (FFA, DAG and oxylipins) are the ones with the most relevant effect, measured as OD fold-change, considering 0.0025mg mL$^{-1}$ concentration versus control. The Y-axis displays the optical density (OD). The X-axis displays the compounds (Statistical significance assessed trough Kruskal Wallis test, p-value <0.05). Horizontal line in each boxplot indicates the median and black dots represent the outlier samples.

its ratio with the linoleic acid could promote virulence [33], whilst DAG associated compounds (e.g. DAG 36:2) lead to establish an appropriate defence response by inducing defence-signalling molecules [33]. As reported elsewhere, oleic acid-derived DOX-oxylipins are instrumental in moderating the lifestyle of another opportunistic pathogen: *P. aeruginosa* [24]. Concerning LOX-derived products, in animal systems, 9-oxylipins (e.g. 9-HODE) may display a pro-inflammatory effect, whereas 13-oxylipins (e.g. 13-HODE) has an anti-inflammatory one [35]. Some authors suggested using 13- to 9-oxylipins ratio as a marker of the inflammatory status in murine models [36]. 9- and 13-oxylipins play a similar, apparently antagonistic role in fungal–plant interaction [37]. Our results show that 18:1 DOX-oxylipins (e.g. 7,10-diHOME) hold back biofilm formation, while LOX-derived 9-oxylipins (e.g. 9-HODE) stimulate biofilm formation. Regarding a possible antagonistic role for 9- and 13-oxylipins, we noticed a similar antagonistic effect in biofilming (i.e. 9- and 13-HOTRE).

Our hypothesis provides the following scenario: plants produce oxylipins in response to several stresses, including which pathogenic insult [38]. Oxylipins, other than functioning as plant defence signals [39], may display an effect toward pathogens in modulating their lifestyle, or a biocidal one [40, 41]. Notably, LOX-derived products could interact with the membranes of pathogens altering their organization and disturbing cell growth [42]. Further studies are needed to clarify how Xf can adapt to oxylipins-modulated plant stress response and exploit it to trigger its biofilming.

We highlight for the first time the different accumulation of FFA, oxylipins and DAG in OQDS+ and Xf+ olive trees and we suggest their role in the modulation of Xf subsp. *pauca* biofilming. We encourage researchers to investigate oxylipins as new targets for the development of treatments for OQDS in line with the Roper team suggestion: "design therapeutics that target the dispersal state or encourage adhesion to lock the bacterial population into its self-limited state permanently" [14].

A more straightforward outcome of our results leads to exploit lipids as markers for developing diagnostic methods based on both destructive strategies, such as molecular tests (lipids related genes) and/or MS spectrometry analysis, and on non-destructive tests such as volatilome of Volatile Organic Compounds (VOCs) derived lipids, as recently suggested [43,44].

## Supporting information

**S1 Fig. Summary of OQDS and Xf real-time PCR screen results.** A) Amplification curves of real-time PCR of OQDS- samples and B) of OQDS+ samples following the protocol published by Harper and colleague [29].
(DOCX)

**S2 Fig. Volcano plot analysis of 524 compounds on positive ion scan, in Xf+ and Xf- samples.** The x-axis shows the fold change to indicate the variation in abundance of compounds present in Xf+ compared to Xf-. The compounds on the right side were more abundant, whereas those on the left side were less abundant in Xf+ condition. Entities that satisfied the fold change and the *P*-value cut-off of 1.5 and 0.05, respectively, are marked in red.
(DOCX)

**S3 Fig. Impact on growth development at different concentrations of individually added lipids (DAG 36:2, DAG 36:4, oleic and linoleic acids, 7,10-DiHOME and 7,10-DiHOME and 10-HOME mix) to *in vitro* cultures of *Xylella fastidiosa*.** Impact is expressed as OD600 fold-change compared to the control, the effect of the corresponding amounts of ethanol employed to dissolve the compounds has been subtracted. Concentrations are expressed in molarity. The effect of ethanol at equivalent and higher molarities is plotted as a reference. Tested compounds were evaluated through Kruskal Wallis test and classified by Fisher LSD post-hoc test with Bonferroni correction (p-value<0,05).
(DOCX)

**S4 Fig. Impact on biofilm production at different concentrations of individually added LOX-oxylipins to *in vitro* cultures of *Xylella fastidiosa* subsp. *pauca* strain De Donno (CFBP 8402).** Impact is expressed as OD595 fold-change compared to the control. Concentrations are expressed in molarity. The effect of ethanol at equivalent and higher molarities is plotted as a reference. Tested compounds were evaluated through Kruskal Wallis test and classified by Fisher LSD post-hoc test with Bonferroni correction (p-value<0,05).
(DOCX)

**S5 Fig. Impact on biofilm production at different concentrations of individually added lipids (DAG 36:2, DAG 36:4, oleic and linoleic acid, 7,10-DiHOME and 7,10-DiHOME and 10-HOME mix) to *in vitro* cultures of *Xylella fastidiosa* subsp. *pauca* strain De Donno (CFBP 8402).** Impact is expressed as OD595 fold-change compared to the control, the effect of the corresponding amounts of ethanol employed to dissolve the compounds has been subtracted. Concentrations are expressed in molarity. The effect of ethanol at equivalent and higher molarities is plotted as a reference. Tested compounds were evaluated through Kruskal Wallis test and classified by Fisher LSD post-hoc test with Bonferroni correction (p-value<0,05).
(DOCX)

**S6 Fig. Impact on growth development at different concentrations of individually added LOX-oxylipins to *in vitro* cultures of *Xylella fastidiosa*.** Impact is expressed as OD600 fold-change compared to the control, the effect of the corresponding amounts of ethanol employed to dissolve the compounds has been subtracted. Concentrations are expressed in molarity. The effect of ethanol at equivalent and higher molarities is plotted as a reference. Tested compounds were evaluated through Kruskal Wallis test and classified by Fisher LSD post-hoc test with Bonferroni correction (p-value<0,05).
(DOCX)

**S1 Table. Analysed lipid entities and relative parameters for Multiple Reaction Monitoring (MRM) experiments and Single Ion Monitoring (SIM).** A) MRM analysis method for oxylipins; B) MRM method for phospholipids, glycerolipids, ornitholipids, bactophenols; C) SIM method for free fatty acids.
(DOCX)

**S2 Table. Compound identification from LC-TOF untargeted analysis.** Each row provides the compound putative identification and values explaining differences between Xf-positive and Xf-negative samples, i.e.: p-value, corrected p-value, Fold Change of Xf-positive versus Xf-negative, abundance difference of Xf-positive versus Xf-negative in raw and log2 form. Compounds highlighted in yellow were fragmented and characterized through product ion experiments.
(DOCX)

**S3 Table. Normalized peak areas of SIM and MRM analyzed lipid entities.** Compounds' peak areas were divided by their internal standard's peak area and then by the maximum value of internal standard peak area. For each compound the fold-change of Xf+ samples versus Xf- samples was calculated, as such the p-value from Student T-test. Compounds with a p-value <0.05 are represented on the upper part of the volcano plot, above the red line corresponding to $-\log_{10}(0.05)$.
(DOCX)

## Author Contributions

**Conceptualization:** Valeria Scala, Marco Zaccaria, Massimo Reverberi.

**Data curation:** Valeria Scala, Nicoletta Pucci, Manuel Salustri, Marco Scortichini, Marco Zaccaria.

**Formal analysis:** Nicoletta Pucci, Babak Momeni.

**Funding acquisition:** Marco Scortichini, Babak Momeni, Massimo Reverberi, Stefania Loreti.

**Investigation:** Valeria Scala, Vanessa Modesti, Alessia L'Aurora.

**Methodology:** Nicoletta Pucci, Manuel Salustri, Vanessa Modesti, Alessia L'Aurora.

**Project administration:** Stefania Loreti.

**Resources:** Marco Scortichini, Stefania Loreti.

**Software:** Manuel Salustri.

**Supervision:** Valeria Scala, Marco Scortichini, Babak Momeni, Massimo Reverberi, Stefania Loreti.

**Validation:** Stefania Loreti.

**Writing – original draft:** Valeria Scala, Massimo Reverberi.

**Writing – review & editing:** Valeria Scala, Nicoletta Pucci, Marco Scortichini, Marco Zaccaria, Babak Momeni, Massimo Reverberi, Stefania Loreti.

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
