## [Decision Letter · Decision Letter 0]

24 Feb 2020

PONE-D-19-35039

Xylella fastidiosa and plant produced lipids control the switch adhesive versus non-adhesive state and viceversa

PLOS ONE

Dear Dr. Reverberi,

Thank you for submitting your manuscript to PLOS ONE. After careful consideration, we feel that it has merit but does not fully meet PLOS ONE’s publication criteria as it currently stands. Therefore, we invite you to submit a revised version of the manuscript that addresses the points raised during the review process.

We would appreciate receiving your revised manuscript by Apr 09 2020 11:59PM. To enhance the reproducibility of your results, we recommend that if applicable you deposit your laboratory protocols in protocols.io, where a protocol can be assigned its own identifier (DOI) such that it can be cited independently in the future. For instructions see: http://journals.plos.org/plosone/s/submission-guidelines#loc-laboratory-protocols

We look forward to receiving your revised manuscript.

Kind regards,

Chih-Horng Kuo, Ph.D.

Academic Editor

PLOS ONE

Journal Requirements:

4. Please ensure that you refer to Figure 5 in your text as, if accepted, production will need this reference to link the reader to the figure.

Reviewers' comments:

Reviewer's Responses to Questions

**Comments to the Author**

1. Is the manuscript technically sound, and do the data support the conclusions?

Reviewer #1: Partly

Reviewer #2: Partly

2. Has the statistical analysis been performed appropriately and rigorously? 

Reviewer #1: Yes

Reviewer #2: No

3. Have the authors made all data underlying the findings in their manuscript fully available?

Reviewer #1: Yes

Reviewer #2: No

4. Is the manuscript presented in an intelligible fashion and written in standard English?

Reviewer #1: Yes

Reviewer #2: Yes

5. Review Comments to the Author

Reviewer #1: This study determined the lipidomic profile of Xf-infected versus uninfected Olea europaea (symptomatic versus symptomless for Olive Quick Decline Syndrome OQDS). The authors selected ten lipidic compounds differentially detected in the Xf-plants and study their effect on Xf biofilm formation and growth in vitro. They show that some of these compounds have an effect (positive or negative) on biofilm formation in vitro. This work is important since it provide initial evidence of the existence of a Xf-plant host cross-talk mediated by lipid molecules. However, the main limitation of this work is that it lacks studies in vivo, which would have greatly support the cross-talk hypothesis. For example, some experiment co-inoculating olive leaves with Xf and the lipidic compound to measure the effect on symptoms and biofilm in vivo. Fluorescent Xf can be used to observe biofilm in vivo into the xyleme vessels.

At the end on the Introduction the authors state: “Importantly, we found that these lipids can be part of a novel Xf quorum sensing control system during its pathogenic relationship with the host”. Although this claim might be correct for some of the lipid molecules studied, it is weakly supported by the experimental evidence presented in the manuscript. The consensus definition of bacterial quorum sensing (QS) involves self-produced extracellular chemical signals, which can accumulate in a local environment to levels that are required to activate transcription of specific effector genes. The authors neither present experimental evidence showing that the lipidic compounds accumulate in a cell density dependent fashion, nor they show that this molecules function as signals to induce a specific subset of effector genes. The evidence provided to this respect is indirect and tangential. In addition, some of the molecules are produced by the plant, which cannot fulfill the definition of QS.

This reviewer thinks that for the paper to be accepted either, 1) the authors should provide more evidence (preferentially in vivo) of the bacterial-host cross-talk and of the QS nature of some of the lipid molecules produced by Xf, or 2) the authors should refocus the manuscript by making more emphasis in the importance of lipidomic determination of Xf-infected vs uninfected olive plants and by unbolding the claims regarding the QS and cross-talk hypotheses. The QS and cross-talk hypotheses should be stated more cautiously and hypothetical. The value of the differential lipidome can be exploited in the discussion as a way to diagnose Xf in olive crops perhaps.

Minor issues:

• Olive Quick Decline Syndrome (OQDS) should be defined the first time mentioned in the abstract.

• The figures can be improved by enlarging some difficult to read fonts.

• In the discussion avoid to repeat information already provided in the Introduction and Results sections.

Reviewer #2: Comments to author are attached.

The study showed streamlined methods in lipid profiling of Xf infected plants from the field, which provides insights into the disease in the field. However, the basis for doing bacterial growth assays was not well elaborated and it was unclear why certain assays and conditions were used. Parameters and findings from the lipidomic analysis were not well interpreted. I suggest the authors address these points through data analysis and assessment of bacterial infection/burden/biofilm formation in the Xf+ samples before resubmission.

6. PLOS authors have the option to publish the peer review history of their article (what does this mean?). If published, this will include your full peer review and any attached files.

Reviewer #1: No

Reviewer #2: Yes: Mengyao Niu

---

## [Author Response · Author response to Decision Letter 0]

30 Mar 2020

Answer to Editor

Dear, 

We have inserted and corrected all the information and suggestions proposed by the reviewers. Specifically, concerning suggestion proposed by reviewer#1 due to lab closure for covid-19 restrictions we were not able to perform the in planta experiments suggested and we choose to reformulate completely our paper following his/her suggestion as second option. Regarding the reviewer#2 we answered all his/her criticisms/suggestions and inserted novel supplementary material. We removed the sentence regarding the not shown data as suggested by your editorial office.

Answer to Reviewer #1: This study determined the lipidomic profile of Xf-infected versus uninfected Olea europaea (symptomatic versus symptomless for Olive Quick Decline Syndrome OQDS). The authors selected ten lipidic compounds differentially detected in the Xf-plants and study their effect on Xf biofilm formation and growth in vitro. They show that some of these compounds have an effect (positive or negative) on biofilm formation in vitro. This work is important since it provide initial evidence of the existence of a Xf-plant host cross-talk mediated by lipid molecules. However, the main limitation of this work is that it lacks studies in vivo, which would have greatly support the cross-talk hypothesis. For example, some experiment co-inoculating olive leaves with Xf and the lipidic compound to measure the effect on symptoms and biofilm in vivo. Fluorescent Xf can be used to observe biofilm in vivo into the xyleme vessels.

At the end on the Introduction the authors state: “Importantly, we found that these lipids can be part of a novel Xf quorum sensing control system during its pathogenic relationship with the host”. Although this claim might be correct for some of the lipid molecules studied, it is weakly supported by the experimental evidence presented in the manuscript. The consensus definition of bacterial quorum sensing (QS) involves self-produced extracellular chemical signals, which can accumulate in a local environment to levels that are required to activate transcription of specific effector genes. The authors neither present experimental evidence showing that the lipidic compounds accumulate in a cell density dependent fashion, nor they show that this molecules function as signals to induce a specific subset of effector genes. The evidence provided to this respect is indirect and tangential. In addition, some of the molecules are produced by the plant, which cannot fulfill the definition of QS.

This reviewer thinks that for the paper to be accepted either, 1) the authors should provide more evidence (preferentially in vivo) of the bacterial-host cross-talk and of the QS nature of some of the lipid molecules produced by Xf, or 2) the authors should refocus the manuscript by making more emphasis in the importance of lipidomic determination of Xf-infected vs uninfected olive plants and by unbolding the claims regarding the QS and cross-talk hypotheses. The QS and cross-talk hypotheses should be stated more cautiously and hypothetical. The value of the differential lipidome can be exploited in the discussion as a way to diagnose Xf in olive crops perhaps.

Thank you for your kind suggestion. Since we are currently working on a set of mutant strains showing defective for oxylipin-forming enzymes, we accept your suggestions and will include another set of data in a following paper still underway. We agree with you that, at this stage, we cannot fully support the hypothesis proposed, so we we eliminated or “unboldened” it in the revised version of this paper.

Minor issues:

• Olive Quick Decline Syndrome (OQDS) should be defined the first time mentioned in the abstract. 

Thanks for your suggestion. We have defined OQDS in the abstract

• The figures can be improved by enlarging some difficult to read fonts.

The font of the boxplot figures have been enlarged. 

• In the discussion avoid to repeat information already provided in the Introduction and Results sections.

The discussion has been completely revised avoiding repetitions from the introduction and results.

Answer to Reviewer#2

Summary: In the current submitted manuscript, the authors described their work in profiling the lipids of Xylella fastidiosa, both from infected olive trees and the uninfected counterparts. Furthermore, they identified free fatty acids derivatives that are differentially represented in these two groups, and some of those inhibit or stimulate the growth of X. fastidiosa. 

Comments: 

Dear reviewer,

replies to your comments are shown in italics

Line 96. It was unclear which part of the plant was collected and how.

The description of how samples collection was performed is now added to the manuscript at L72-77

Line 101. Can authors elaborate further what they mean by “six pools from OQDS+ and six from OQDS-“? Does it mean that 10 samples from each group were pooled together? How many samples (or pools) then were subjected to the lipidomic analysis? 10 samples (derived from 10 different olive tresses) were pooled together and 12 samples were analysed.

Lipidomic analysis was performed on 12 pools for a total of 120 plants (60 infected and 60 non-infected); each pool was made putting together samples harvested from 10 plants. The sentence has been modified accordingly at L78-80

Line 156. Are most of the symptomless samples free of Xf, and are the OQDS+ samples mostly infected by Xf? Please provide a summary of OQDS and Xf PCR screen results. 

The real time PCR results are reported in this answer and, now, in S1 Figure. The figure S1A shows the results of OQDS- samples, the figure S1B the results of OQDS+.

A)

A) B)

Samples were amplified following the protocol published by Harper and colleague [29].

Line 159. The authors described two principle components in Figure 2-do both PC’s differentiate the Xf+ and Xf- samples? Please clarify. 

The PC1 distinguish Xf+ from Xf-. PC2 is more related to the type of lipids present as highlighted in the corresponding loading plot. From this, we can suggest that specific type of lipids are hallmarks of infection (e.g. oxylipins), some others are probably more related to specificities in lipid composition of the different pools.

Line 162. Please elaborate the reasons for selecting the 90 compounds for further analysis. What previous results specifically were the authors referring to? Please also provide a table with fold change and p-values listed for these 90 compounds. 

The select compounds emerged from the previous dataset published in Scala et al., 2018 doi https://doi.org/10.3389/fmicb.2018.01839 [30], and from selecting compounds on the upper right and left box within the volcano plot of Figure 1 straight from mass profiler professional output. Then, on these 90 compounds we performed product ion fragmentation to confirm the ToF analysis. On those positively identified (~80%) we performed MRM and SIM analysis to quantify them. Within this dataset (Table S2), only 25 were revealed within our samples (both Xf- and Xf+) and consequently quantified by MRM (now shown in Table S3). 10 out of 25 resulted significantly different between Xf+ and Xf- samples (T Test and volcano plot in Table S3) and were used for building up the boxplots of Figure 3.

Fig 1. Statistical corrections be taken into account for multivariant analysis. A more stringent cut-off should be applied. 

We worked on default settings of the mass profiler version (Cut-offs were 2 fold-change and 0,05 p-value) working downstream, but knowing that this analysis could be more stringent we chose almost half of the emerged compounds from the volcano plot and specifically those presenting the highest FC and corrected P value. 

Line 165. In Fig 3 the authors showed 10 lipids that are differentially abundant between the Xf+ and Xf- samples. It was unclear why these 10 lipids were chosen, so please elaborate further. Are they the most different ones? Did these show the biggest fold change? 

Among the set of 90 targets, twe chose the most different between the Xf+ and Xf- samples. The 10 compounds are the statistically significant ones (T-test, p-value<0,05; Table S3) among targeted analysis tested compounds (listed in MRM table in Material and Methods), as also stated above at answer at lane 162 question.

Line 182, Fig 3. The authors said “The Y-axis shows the normalized relative abundance” though it is not clear how the normalization was done, and the axes appeared arbitrary. Please elaborate how these numbers were calculated. 

Target compounds’ peak areas were divided by their internal standard’s peak area and then by the maximum value of internal standard peak area

Line 188. Even though other studies have shown SEMs of Xylella fastidiosa infecting other plants and forming biofilm (Newman et al., 2004), it was not confirmed that they do the same in the host being studied here. Can the authors confirm with a subset of Xf+ samples that biofilm was indeed formed? This will provide basis for linking lipid profiles with planktonic/biofilm growth. 

Unfortunately, we cannot confirm the biofilm formation in Xf+ olive samples, but we are supported by Cardinale et al 2018 [34] who demonstrated that Xf form biofilm into symptomatic olive trees cultivar (“Ogliarola di Lecce”). The cultivar used in this study was the same we employed. Cardinale et al 2018 evaluated the role of bacterial aggregates in vessel occlusions and quantified the level of infection and vessel occlusion in both petioles and branches of naturally infected and non-infected olive trees. The authors demonstrated that symptomatic petioles showed colonization by X. fastidiosa, and the vessels appeared completely occluded by a biofilm containing bacterial cells and extracellular matrix. Thus, Cardinale and colleagues point out the primary role of the pathogen in olive vessel occlusions and underline the bacteria role in causing vessel occlusion in leaves. The paper is now cited in ref section [34]. 

Line 192. It was not shown that 7,10-diHOME were different between the Xf+ and Xf- samples, so why was this tested? Why was a mixture of 7,10-diHOME and 10-HOME tested? 

The oleic-derived oxylipins present in figure 3 is the 10-HpOME that is a relatively unstable product and thus difficult to provide in vitro. Following the study of Martinez et al. 2016 [24], 10-HpOME is readily converted to 7,10-diHOME and 10-HOME. These products are commercially unavailable, and we asked the Martinez group to provide them and they kindly send us these oxylipins. Unfortunately, 10-HOME is in mixture with 7,10-diHOME and the 7,10-diHOME was pure. We inserted them in the in vitro tests since they are, as indicated in Martinez et al 2016 and 2018 [24, 25], part of the oleic acid oxylipin pathway and a quorum-sensing molecule in Pseudomonas aeruginosa; furthermore, in Scala et al. 2018 [30] we found 7,10-diHOME in Xf, thus in this manuscript we aim at verifying if this compound could act in Xf similarly to P. aeruginosa. 

Minor point here: 9-HODE induced biofilm formation was shown in Fig 5, not Fig 4. 

Concerning Fig. 5, we suppose it is due to the consequent numeration generated by the PloS submitting system. Thus, the boxplots are reported in the second page of the figure 4.

Line196. Can authors provide statistical support for the effect of these compounds shown in Fig S2? 

Tested compounds were evaluated through Kruskal Wallis test and classified by Fisher LSD post-hoc test with Bonferroni correction (p-value<0,05). 

Decision: 

The study showed streamlined methods in lipid profiling of Xf infected plants from the field, which provides insights into the disease in the field. However, the basis for doing bacterial growth assays was not well elaborated and it was unclear why certain assays and conditions were used. Parameters and findings from the lipidomic analysis were not well interpreted. I suggest the authors address these points through data analysis and assessment of bacterial infection/burden/biofilm formation in the Xf+ samples before resubmission. 

- The basis for bacteria growth assays was elaborated in relation to the results of Martinez et al., 2016 [24] whereas the growing conditions were chosen in relation to the information reported in Scala et al 2018 [30]. In relation to this issue, more information were now added in the text. 

- The interpretation of parameters and finding of lipidomic analysis were explained in the replies to reviewer comments (see above). Therefore, the interpretation of lipidomic results was conduct following the available literature on X. fastidiosa and on laboratory experience in lipidomics in plant disease and by closely following the outputs of mass profiler professional provide by Agilent technologies. 

- The bacterial infection/burden/biofilm formation in the Xf+ samples is assessed in the replies above and in the text

---

## [Decision Letter · Decision Letter 1]

28 Apr 2020

Xylella fastidiosa subsp. pauca and olive produced lipids control the switch adhesive versus non-adhesive state and viceversa

PONE-D-19-35039R1

Dear Dr. Reverberi,

We are pleased to inform you that your manuscript has been judged scientifically suitable for publication and will be formally accepted for publication once it complies with all outstanding technical requirements.

With kind regards,

Chih-Horng Kuo, Ph.D.

Academic Editor

PLOS ONE

Additional Editor Comments (optional):

Dear authors,

Congratulations on the nice revision. Both reviewers are satisfied with the changes and I agree with the assessment. Reviewer #2 have provided some additional comments in a separate file. I believe that these are pretty minor and could be corrected prior to production, there is no need for this manuscript to go through another round of scientific review.

Best, CH

Reviewers' comments:

Reviewer's Responses to Questions

**Comments to the Author**

1. If the authors have adequately addressed your comments raised in a previous round of review and you feel that this manuscript is now acceptable for publication, you may indicate that here to bypass the “Comments to the Author” section, enter your conflict of interest statement in the “Confidential to Editor” section, and submit your "Accept" recommendation.

Reviewer #1: All comments have been addressed

Reviewer #2: All comments have been addressed

2. Is the manuscript technically sound, and do the data support the conclusions?

Reviewer #1: Yes

Reviewer #2: Yes

3. Has the statistical analysis been performed appropriately and rigorously? 

Reviewer #1: Yes

Reviewer #2: Yes

4. Have the authors made all data underlying the findings in their manuscript fully available?

Reviewer #1: Yes

Reviewer #2: Yes

5. Is the manuscript presented in an intelligible fashion and written in standard English?

Reviewer #1: Yes

Reviewer #2: (No Response)

6. Review Comments to the Author

Reviewer #1: (No Response)

Reviewer #2: The authors addressed our questions through detailing the rationales behind the study and supplementing data and methods that were previously missing or poorly explained. This overall meets the expectation. I accept this manuscript if the authors address a few points highlighted in attached document.

7. PLOS authors have the option to publish the peer review history of their article (what does this mean?). If published, this will include your full peer review and any attached files.

Reviewer #1: No

Reviewer #2: Yes: Mengyao Niu

---

## [Editor Report · Acceptance letter]

4 May 2020

PONE-D-19-35039R1 

*Xylella fastidiosa* subsp. *pauca* and olive produced lipids moderate the switch adhesive versus non-adhesive state and *viceversa*

Dear Dr. Reverberi:

I am pleased to inform you that your manuscript has been deemed suitable for publication in PLOS ONE. Congratulations! Your manuscript is now with our production department. 

With kind regards,

on behalf of

Dr. Chih-Horng Kuo 

Academic Editor

PLOS ONE